# SELF-IMPROVING LLM AGENTS AT TEST-TIME

## ABSTRACT

One paradigm of language model (LM) fine-tuning relies on creating large training datasets, under the assumption that high quantity and diversity will enable models to generalize to novel tasks after post-training. In practice, gathering large sets of data is inefficient, and training on them is prohibitively expensive; worse, there is no guarantee that the resulting model will handle complex scenarios or generalize better. Moreover, existing techniques rarely assess whether a training sample provides novel information or is redundant with the knowledge already acquired by the model, resulting in unnecessary costs. In this paper, we explore a new test-time self-improvement method to create more effective and generalizable agentic LMs *on-the-fly*. The proposed algorithm can be summarized in three steps: (i) first it identifies the samples that the model struggles with by using an uncertainty function (self-awareness), (ii) then generates similar examples from the detected uncertain samples (self-data augmentation), and (iii) uses these newly generated samples at test-time fine-tuning (self-learning). We study two variants of this approach: *Test-Time Self-Improvement* (TT-SI), where the same model generates additional training examples from its own uncertain cases and then learns from them, and contrast this approach with *Test-Time Distillation* (TT-D), where a stronger model generates similar examples for those same uncertain cases, enabling the student to adapt using distilled supervision. Empirical evaluations across different agent benchmarks demonstrate that TT-SI surpasses other standard learning methods with +5.36% absolute gain in average accuracy, yet trains using $68\times$ less training samples and TT-D further improves performance in harder scenarios that require diverse training signals. Our findings highlight the promise of TT-SI and limitations in current learning frameworks regarding cost and generalizability, demonstrating the potential of self-evolving LMs at test-time as a new paradigm for building more capable agents on complex scenarios.

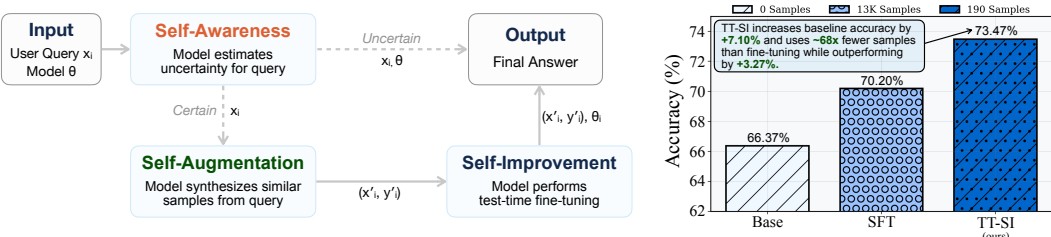

Figure 1: **Overview of the Test-Time Self-Improvement (TT-SI) framework. (Left)** The proposed framework enables *on-the-fly* adaptation by targeting uncertain test instances. It consists of three steps: **(1) Self-Awareness:** An **Uncertainty Estimator (H)** identifies challenging samples. **(2) Self-Data Augmentation:** For each identified uncertain sample, one similar variant is automatically generated using **Data Synthesis Function (G)**, expanding the local decision boundary with minimal overhead. **(3) Self-Improvement:** A lightweight update is performed via **Test-Time Fine-tuning (T)** on these generated samples, using only *one additional training instance per uncertain case*. **(Right)** Comparison of results for standard prompting (Base), supervised fine-tuning (SFT), and TT-SI (ours) on the SealTool benchmark, where both training and test splits are available. TT-SI improves baseline accuracy by +7.1% during inference and surpasses standard SFT by +3.3% with $68\times$ fewer training samples (190 vs. 13k).

## 1 INTRODUCTION

Recent progress in language model (LM) post-training has shown promising results across a wide range of tasks (Kumar et al., 2025) by equipping these models with explicit knowledge (Grattafiori et al., 2024; Yang et al., 2025), reasoning (Zelikman et al., 2022; Guo et al., 2025), and agentic capabilities (Zeng et al., 2024; Chen et al., 2024). These systems are typically trained to approximate an unknown mapping $\mathcal{F}_\theta : \mathcal{X} \to \mathcal{Y}$ from large-scale collections of input–output pairs $(x_i, y_i)$, where $\mathcal{X}$ denotes the inputs and $\mathcal{Y}$ denotes their corresponding desired targets. In this approach, a single function $F_\theta$ attempts to cover all the relevant knowledge and generalization capability from a single dataset $X$, implicitly assuming that the dataset has sufficient quality, diversity, and scale to effectively learn diverse tasks. However, this learning paradigm can remain narrow and inefficient compared to actual human learning (Mitchell et al., 2018).

In contrast, humans take advantage of their background experience (similar to the pretraining stage of LMs) and exhibit remarkable efficiency during learning, often guided by self-regulated learning principles (Zimmerman, 2002) where individuals actively seek and learn from strategically informative demonstrations (Nelson, 1990). For example, consider a student who is preparing for a college entrance exam after years of coursework. Engaging in metacognitive reflection (Flavell, 1979), the student can either broadly practice questions on various topics (e.g., algebra, history, chemistry) or strategically identify gaps in their knowledge (**self-awareness**), collect targeted questions addressing these specific deficiencies (**self-augmentation**), and practice them repeatedly (**self-learning**). Clearly, the second strategy is more effective and explicitly improves the required knowledge (see Appendix B for another example).

The same inefficiency is evident in the standard LM agent fine-tuning paradigms, which train the agentic models to inductively learn general rules from training data to be applied to new, unseen test instances such as tool use or other complex agentic tasks. It involves gathering large-scale training datasets (Ouyang et al., 2022; Wang et al., 2023; Zeng et al., 2024) (either human-curated or LLM-synthesized) and fine-tuning models on these datasets (Grattafiori et al., 2024; Zeng et al., 2024; Acikgoz et al., 2025). However, constructing these datasets is costly, often requiring days to weeks of computation and manual labor, and still provides no guarantee of effective performance and generalization after fine-tuning. Moreover, this approach implicitly assumes that models must process every sample, without considering if certain examples are redundant or already known by the LM. Based on these deficiencies, a key open question is *whether models can be trained to acquire new skills more efficiently, without relying on exhaustive datasets or processing large amounts of redundant information.*

Motivated by local and transductive learning (Bottou & Vapnik, 1992; Joachims, 1999) with recent advances in test-time fine-tuning (Akyürek et al., 2025), we investigate a simple, yet powerful, instance-specific self-improvement algorithm that adapts agents *on-the-fly* to each downstream task at test-time (Figure 1). The proposed algorithm first identifies the most informative and challenging samples while discarding mastered or redundant ones, guided by the designed **Uncertainty Estimator** (**H**), which reflects *self-awareness*. For each retained "necessary" test instance, the model synthesizes a set of distributionally similar samples with **Data Synthesis Function** (**G**) as *self-augmentation* and performs temporary gradient updates with **Test-Time Fine-tuning** (**T**) through *self-learning* on these instances. We explore two different variants of our approach: Test-Time Self-Improvement (TT-SI), where the model trains on self-generated samples using parameter efficient fine-tuning techniques (PEFT) (Hu et al., 2022), and Test-Time Distillation (TT-D) where adaptation is guided by supervision from samples synthesized by a more capable teacher model.

We demonstrate that *test-time self-improvement* enables agents to adapt on-the-fly by leveraging their own uncertain predictions. With only a *single synthesized training instance* per test case, TT-SI shows consistent absolute accuracy gains across three challenging agent benchmarks: +6.05% on NexusRaven, +5.66% on SealTool, and +4.26% on API-Bank. These improvements highlight that even minimal, uncertainty-guided adaptation can substantially boost performance during inference. Moreover, TT-D further extends these gains in complex, context-heavy scenarios (e.g., multi-turn conversations). Compared to standard supervised fine-tuning (SFT), TT-SI surpasses accuracy on SealTool while using $68\times$ fewer samples, underscoring efficiency without compromising effectiveness. We find that, when training is infeasible, TT-SI with in-context learning (ICL) offers a fast, training-free alternative, outperforming other standard learning methods in similar condition.

Concretely, our main findings and contributions can be summarized as follows:

- We propose a three-stage algorithm for *test-time self-improvement*, motivated by human learning theories: (i) identify uncertain samples via a novel uncertainty estimator, (ii) generate new training instances similar to these samples, and (iii) update the model online.
- We conduct a systematic empirical study of two variants, TT-SI and TT-D, analyzing key components such as the choice of uncertainty estimator, learning method at test time, scaling of generated samples, and other parameter effects.
- We validate that agentic LMs can self-improve during inference, even from a single training instance, and show that our framework outperforms standard inductive learning approaches, achieving significant gains with orders-of-magnitude less compute through both test-time ICL and test-time fine-tuning.

Overall, our work pioneers a novel algorithmic framework for agent learning, inspired by human-like lifelong adaptation, seamlessly integrating self-awareness, targeted self-generated data, and iterative self-training to enable continuous self-improvement. We propose that, with an optimal uncertainty estimator to identify weaknesses, precise data synthesis to address them, and focused continual training, agents can continually advance toward mastering increasingly complex and diverse tasks. Thus, this study opens new research directions along these three interconnected paths, **rethinking how LLM agents learn, adapt, and generalize.**

## 2 PRELIMINARIES

### 2.1 FUNDAMENTAL ISSUES IN INDUCTIVE FINE-TUNING

The standard post-training paradigm separates training and testing: models are trained by *inductively* extracting generalizable patterns from data and subsequently evaluated on new, possibly unseen examples (Vapnik, 1999; LeCun et al., 2015; Zhang et al., 2024). Current approaches for training LMs largely follow this paradigm, relying on large-scale post-training datasets. Formally, these datasets are denoted as $\mathcal{D}_{\text{train}} = \{(x_i, y_i)\}_{i=1}^{N}$, consist of $N$ samples assumed to be independently and identically distributed (i.i.d.) according to a *training* distribution $\mathcal{P}_{\text{train}}(\mathcal{X}, \mathcal{Y})$.

Here, $x_i \in \mathcal{X}$ represents an input (e.g., a task query) and $y_i \in \mathcal{Y}$ is its corresponding desired output (e.g., a sequence of actions for an agent). The objective is to find the parameters $\theta$ of a mapping function $\mathcal{F}_\theta : \mathcal{X} \to \mathcal{Y}$, representing the agent, that minimize the empirical risk on the training data: $\hat{\mathcal{L}}_{\text{train}}(\theta) = \frac{1}{N} \sum_{i=1}^{N} \ell(\mathcal{F}_\theta(x_i), y_i)$, where $\ell$ is a predefined loss function. The foundational assumption is that if $N$ is sufficiently large and $\mathcal{D}_{\text{train}}$ is diverse enough, the learned model $\mathcal{F}_\theta$ will generalize effectively to new, unseen inputs $x$ drawn from a *test* distribution $\mathcal{P}_{\text{test}}(\mathcal{X})$. However, this prevailing paradigm is beset by several fundamental issues:

- **Distributional Shift:** Test distributions $\mathcal{P}_{\text{test}}$ often differ from the training distribution $\mathcal{P}_{\text{train}}$ (i.e., $\mathcal{P}_{\text{test}} \neq \mathcal{P}_{\text{train}}$). This means the empirical risk $\hat{\mathcal{L}}_{\text{train}}(\theta)$ provides a misleading picture of the true test risk $\mathcal{L}_{\text{test}}(\theta) = \mathbb{E}_{(x,y)\sim\mathcal{P}_{\text{test}}}[\ell(\mathcal{F}_\theta(x), y)]$, which in turn impairs the model's generalization to novel or complex scenarios (Liu et al., 2021).
- **Computation Cost:** The reliance on extremely large training datasets $\mathcal{D}_{\text{train}}$ (with $N \gg 10^4$ samples) leads to substantial annotation and computational costs, both scaling with $N$, rendering agent development prohibitively expensive (Mirzasoleiman et al., 2020; Mindermann et al., 2022).
- **Redundancy and Inefficient Use of Information:** Treating all $N$ training samples $(x_i, y_i)$ in $\mathcal{D}_{\text{train}}$ as equally informative is inefficient, as the number of truly effective samples $N_{\text{eff}}$ is often much smaller than $N$ (i.e., $N_{\text{eff}} \ll N$) (Zhou et al., 2023). Processing redundant or already mastered examples wastes computational effort and can even restrict generalization, particularly for inputs from the long tail of the data distribution or adversarial examples, which current agents struggle with (Settles, 2009; Sorscher et al., 2022).
- **Catastrophic Forgetting and Model Churn:** Standard fine-tuning for LMs often suffer from catastrophic forgetting (Luo et al., 2025), where fine-tuning a model on a new task inadvertently degrades its performance on previously acquired skills. Moreover, the rapid release of new and more capable LLMs (Grattafiori et al., 2024; Yang et al., 2025) necessitates a continuous and costly re-training cycle, as the entire fine-tuning process must be repeated on $\mathcal{D}_{\text{train}}$ to leverage the increased knowledge and reasoning abilities of each new base model on the downstream task.

These limitations motivate a new post-training paradigm grounded in *transductive* and *local learning* principles, which adapts the model on-the-fly by identifying and training only on the most informative samples drawn from the test distribution $\mathcal{P}_{\text{test}}$.

## 2.2 TEST-TIME TRAINING

Test-time training (TTT) performs small, ephemeral parameter updates during inference, conditioning the model on the current input and thus partially collapsing the train–test boundary (Sun, 2023). The idea traces to local and transductive learning, where hypotheses are adapted after observing test inputs (Bottou & Vapnik, 1992; Joachims, 1999). In deep learning, Sun et al. (2020) showed that a simple self-supervised TTT objective can improve the robustness of image classifiers under distribution shift. In LLMs, TTT is comparatively nascent: Hardt & Sun (2024) fine-tune on retrieved nearest neighbors to reduce perplexity, and SIFT (Hübotter et al., 2025) actively selects diverse, informative neighbors to limit redundancy. Closest to our setting, Akyürek et al. (2025) apply rule-based linear transformations to in-context test examples in ARC to get additional test-time training data. However, these approaches either target perplexity rather than general reasoning tasks, assume access to high-quality neighbors or in-context exemplars. Our work instead selects informative test instances, generates and filters training signals on-the-fly, yielding improvements on challenging agent benchmarks. To the best of our knowledge, this is the first language generation–based test-time fine-tuning method applied to LLM-based agents. Further details on prior work in LLM and agent post-training, and how our work differs, are provided in Appendix Section C.

## 3 METHOD

We introduce a test-time self-improvement framework designed to enable agents to learn from challenging instances on-the-fly by integrating three key components, as shown in Algorithm 1:

- **Self-Awareness: Uncertainty Estimator (H)** identifies inputs $x_i$ at inference-time which the agent is uncertain on, ensuring adaptation focuses only on informative, challenging cases (Section 3.1).
- **Self-Augmentation: Data Synthesis Function (G)** generates a set of $K$ new samples ($\mathcal{D}'_i$) that are closely related synthetic examples, generated based on the uncertain input $x_i$ (Section 3.2).
- **Self-Learning: Test-Time Fine-tuning (T)** temporarily updates the agent's parameters ($\theta$) on the targeted synthetic data ($\mathcal{D}'_i$) (Section 3.3).

In the following subsections, we detail each component one by one, first by providing formal definitions followed by their algorithmic specifics.

### 3.1 SELF-AWARE SAMPLE SELECTION AT TEST TIME

This section details our approach for identifying and selecting data samples for which the model $\mathcal{M}$ exhibits high uncertainty during inference. We posit that such samples are more likely to be challenging or error-prone, and are thus particularly informative for further learning.

**Definition** Given a task with inputs $x_i$, we define **Uncertainty Estimator (H)** that estimates the model's confidence score ($\mathcal{C}$) for each candidate action $a_1, ..., a_n \in \mathcal{A}$ available to the model $\mathcal{M}$ in its environment (e.g., available API calls). For each input $x_i$ and candidate action $a_n$, the confidence is computed as:

$$\mathcal{C}_i = \mathbf{H}(x_i, a_n, \mathcal{M}) \tag{1}$$

This estimation is performed without access to ground-truth labels $y_i$, ensuring fairness and applicability during inference. A sample $x_i$ is deemed **uncertain** if $\mathcal{C}_i < \tau$ for a user-defined confidence threshold $\tau$. By filtering out high-confidence (i.e., certain) instances, this uncertainty estimation step focuses computational and learning resources for the most informative and challenging questions, thereby enhancing both efficiency and quality.

**Selecting Uncertain Samples** To systematically identify uncertain samples, we implement a *margin-based confidence estimator* using the likelihood distribution generated by the model $\mathcal{M}$ for a given input $x_i$. Given a set of available actions $a_1, a_2, \ldots, a_N$, we first compute the negative log-likelihood (NLL) for each action as:

$$\text{NLL}(a_n|x_i) = -\log P_{\mathcal{M}}(a_n|x_i), \quad \forall n \in 1, 2, \ldots, N. \tag{2}$$

---

**Algorithm 1** Test-Time Self-Improvement Framework

---

**Require:** Test dataset $\mathcal{D}_{\text{test}}$, model $\mathcal{M}$, data generation prompt $\mathcal{P}$, temporary dataset size $K$, initial model parameters $\theta_0$

1: **for** each $x_i \in \mathcal{D}_{\text{test}}$ **do**
2:     **Step 1: Uncertainty Estimator (H)**
3:     **Compute** uncertainty (softmax-difference):
4:         $\ell_n = -\log P_{\mathcal{M}}(a_n|x_i), \quad \forall a_n$   ▷ Negative Log-Likelihood (NLL) for candidate action
5:         $p_n = \frac{\exp(\ell_n - \max_j \ell_j)}{\sum_k \exp(\ell_k - \max_j \ell_j)}$   ▷ Apply Relative Softmax Scoring (RSS) normalization
6:         $u(x_i) = p^{(1)} - p^{(2)}$   ▷ Highest minus second-highest RSS scores
7:     **Step 2: Data Synthesis Function (G)**
8:     **if** $u(x_i) < \tau$ **then**   ▷ Check uncertainty
9:         **Generate** $K$ synthetic samples using LLM:
10:         $\mathcal{D}_i \leftarrow \mathcal{L}_{\text{gen}}(x_i, K)$   ▷ Equation (5)
11:     **Step 3: Test-Time Fine-tuning (T)**
12:         **Learn** temporary model parameters $\theta_i^*$ via LoRA:
13:         $\theta_i^* \leftarrow \arg\min_{\theta_0} \sum_{(x',y') \in \mathcal{D}_i} \ell(\mathcal{M}(x'; \theta_0), y')$   ▷ Equation (7)
14:         **Perform** inference with adapted parameters $\theta_i^*$:
15:         $\hat{y}_i \leftarrow \mathcal{M}(x_i; \theta_i^*)$
16:         **Reset** model parameters:
17:         $\theta_i^* \rightarrow \theta_0$   ▷ Restore original parameters
18:     **else**
19:         **Perform** inference directly:
20:         $\hat{y}_i \leftarrow \mathcal{M}(x_i; \theta_0)$
21:     **end if**

---

However, raw NLL scores are not directly interpretable due to their unbounded nature, limiting their utility in precisely quantifying uncertainty. To address this issue, we apply a *Relative Softmax Scoring (RSS)* mechanism, which transforms these scores into a normalized and interpretable confidence distribution:

$$p^n = \frac{\exp(\ell_n - \max_j \ell_j)}{\sum_{k=1}^{N} \exp(\ell_k - \max_j \ell_j)}, \quad \text{where} \quad \ell_n = -\text{NLL}(a_n \mid x_i). \tag{3}$$

Here, $p^n$ is the RSS confidence score for action $a_n$, and $\ell_n$ denotes the negative log-likelihood score corresponding to $a_n$. The $\max_j \ell_j$ term represents the maximum NLL score among all candidate actions, serving as a numerical stabilizer. To quantify prediction uncertainty, we compute the difference between the highest and second-highest RSS scores, termed the *softmax-difference*. Formally, uncertainty for input $x_i$ is defined as:

$$u(x_i) = p^{(1)} - p^{(2)}, \tag{4}$$

where $p^{(1)}$ and $p^{(2)}$ denote the highest and second-highest RSS scores, respectively. Finally, using a user-defined threshold $\tau$, we select samples exhibiting high uncertainty ($u(x_i) < \tau$), which ensures that subsequent adaptation or analysis efforts are focused on the most ambiguous instances, where the model is likely to benefit most from further information or refinement.

## 3.2 DATA GENERATION STRATEGIES

Once an individual input sample $x_i$ is identified as exhibiting high uncertainty by the model $\mathcal{M}$ (as per the criteria in Section 3.1), our approach triggers an immediate data synthesis process with **Data Synthesis Function (G)**. This section details the methodology for generating new, relevant training data specifically for the uncertain instance at hand. The core idea is to create a focused, temporary dataset on-the-fly, enabling rapid, localized adaptation of the model to address the specific query it found challenging.

### 3.2.1 DATA SYNTHESIS METHOD

**Definition** When an input sample $x_i$ (without ground-truth labels) is processed during inference and flagged as uncertain by **H**, we trigger the synthesis of $K$ new training examples together with the

corresponding labels. **Data Synthesis Function** (**G**) is invoked for this specific uncertain input $x_i$, producing a set of $K$ new input-output pairs following the provided instructions (Figure 6). These synthetic examples, $(x'_{ij}, y'_{ij})_{j=1}^K$, are aimed to be semantically similar to the original uncertain sample $x_i$ while introducing slight variations. In practice, $x_i$ serves as a seed example in the prompt, guiding the generation of $K$ new synthetic training pairs $(x', y')$ that resemble the original input but expand the training signal (Wang et al., 2023).

The **G** is invoked for this specific uncertain input $x_i$, producing a set of $K$ novel input-output pairs following the provided prompt of instructions ($\mathcal{P}$):

$$\mathbf{G} : x_i \to \{(x'_{ij}, y'_{ij})\}_{j=1}^K \tag{5}$$

Here, $K$ is a user-defined hyperparameter dictating the volume of synthetic data generated for the current uncertain instance $x_i$. Each generated pair $(x'ij, y'ij)$ aims to be a plausible instance from the underlying data distribution $P(x, y)$ relevant to $x_i$, specifically targeting the model's region of uncertainty around this input. These $K$ generated pairs immediately form a temporary, query-specific dataset $\mathcal{D}_i$:

$$\mathcal{D}_i = \{(x'_{ij}, y'_{ij})\}_{j=1}^K \tag{6}$$

This dataset $\mathcal{D}_i$ is then used for a localized adaptation of the model parameters $\theta$ before processing subsequent input samples with **Test-Time Fine-tuning** (**T**) as the next step described in the next section (Section 3.3). This iterative process of detection, synthesis, and adaptation is performed for each sample identified as uncertain.

**Generating Samples** The implementation of the synthesis function **G** (Equation (5)), which is triggered for each uncertain sample $x_i$, employs the agent itself for data synthesis ($\mathcal{L}_{\text{gen}}$) as *self-augmentation*. For each generation instance, $\mathcal{L}_{\text{gen}}$ is provided with a carefully hand-crafted prompt $\mathcal{P}$ (See Figure 6), the uncertain input $x_i$ serving as the direct seed (critically, without its corresponding label $y_i$), and a specified number of samples $K$ to generate. The model then produces $K$ new input-output pairs, denoted as $\{(x'_{ij}, y'_{ij})\}_{j=1}^K$. This seed-based generation process, inspired by self-instruction methodologies (Wang et al., 2023), guides $\mathcal{L}_{\text{gen}}$ to produce variants that maintain the core semantic meaning and task relevance of $x_i$ while introducing controlled surface-level variations. By *synthesizing data in this on-the-fly* manner for each uncertain instance, we facilitate targeted and timely model adaptation, aiming to improve performance on precisely the types of queries the model struggles with, as they are encountered.

### 3.3 TEST-TIME FINE-TUNING

Test-Time Fine-Tuning enables parametric models to update their weights temporarily during inference (Sun, 2023), yet this paradigm is largely unexplored for LLMs (Akyürek et al., 2025) and, to the best of our knowledge, has not been applied to agentic tasks. Building on our previous steps, uncertainty detection (Section 3.1) and targeted data synthesis (Section 3.2), we now use **Test-Time Fine-tuning** (**T**) to adapt model $\mathcal{M}$ during inference, using the generated samples $\mathcal{D}_i$ for each uncertain test query $x_i$.

**Definition** Once we got $\mathcal{D}_i$ with Equation (5), we optimize initial parameters ($\theta_0$) to minimize the loss function $\mathcal{L}(\mathcal{D}_i; \theta_0)$, producing temporarily updated parameters $\theta_i$ for the target task prediction. Importantly, after generating predictions, the model is restored to the original parameters $\theta_0$ for the next iteration using sample $x_{i+1}$, thereby creating a specialized prediction model for each *out-of-distribution* sample without permanently altering the base model.

**Test-Time Fine-Tuning** The primary goal of inference-time adaptation is to temporarily adjust the model $\mathcal{M}$'s parameters ($\theta$) to better handle the current uncertain sample $x_i$. This is achieved by fine-tuning the model on the newly synthesized dataset $\mathcal{D}_i = \{(x'_{ij}, y'_{ij})\}_{j=1}^K$. The adaptation involves minimizing a task-specific loss function $\mathcal{L}_{\text{task}}$ over the samples in $\mathcal{D}_i$. For a given self-generated sample $(x'_{ij}, y'_{ij}) \in \mathcal{D}_i$, the loss is computed as $\ell(\mathcal{M}(x'_{ij}; \theta), y'_{ij})$, and the objective for adapting parameters $\theta$ using dataset $\mathcal{D}_i$ is:

$$\theta_i^* = \arg\min_{\theta'} \sum_{(x'_{ij}, y'_{ij}) \in \mathcal{D}_i} \ell(\mathcal{M}(x'_{ij}; \theta'), y'_{ij}) \tag{7}$$

where $\theta_i^*$ represents the adapted parameters for the context of $x_i$. We employ Low-Rank Adaptation (LoRA) (Hu et al., 2022) to ensure computational efficiency during inference-time updates.

# 4 RESULTS

| Inference | Method | NexusRaven | SealTool | API-Bank | Avg. | $\Delta\%$ |
|---|---|---|---|---|---|---|
| Input/Output | w/o TT-SI (Base) | 44.03 | 66.67 | 70.08 | 60.26 | – |
| | w. TT-SI | 50.08 | 72.43 | 74.34 | 65.62 | ↑5.36 |
| | w. TT-D | **52.52** | **75.17** | **77.29** | **68.33** | ↑8.07 |
| Majority Vote | w/o TT-SI (Base) | 46.56 | 69.73 | 73.96 | 63.42 | – |
| | w. TT-SI | 52.20 | **72.93** | 75.68 | 66.94 | ↑3.52 |
| | w. TT-D | **54.53** | 72.25 | **77.56** | **68.11** | ↑4.69 |

Table 1: **Main Results of TT-SI**. Accuracy results of baseline prompting, TT-SI, and TT-D across three agentic benchmarks under two inference settings: *Input/Output* (direct prediction) and *Majority Vote* (self-consistency). $\Delta$ denotes the average absolute improvement over the corresponding baseline without TT-SI and ↑ indicates performance increase.

**Experimental Protocol**  We evaluate our approach on three complementary agent benchmarks. NexusRaven (Srinivasan et al., 2023) is a function-calling benchmark that tests an agent's ability to execute single, nested, and parallel function calls of varying complexity. SealTool (Wu et al., 2024) is a self-instruct dataset for tool learning, measuring precision in tool selection, adherence to output formats, and adaptability across diverse scenarios. API-Bank (Li et al., 2023) evaluates multi-turn user–agent dialogues, requiring agents to track conversational state, make informed tool calls at each turn, and handle realistic conditions such as noisy or incomplete inputs. We use `Qwen2.5-1.5B-Instruct` for main experiments, as its strong performance and small size allow efficient use of limited hardware and demonstrate the potential of compact agentic models (Belcak et al., 2025). To examine scaling and architectural variations, we further include `Qwen2.5-7B-Instruct` ablations. All models are trained with PEFT using LoRA (Hu et al., 2022) on a single NVIDIA A40 GPU. Because our method often follows a small-sample regime (e.g., single-sample training), higher deviation is expected; thus, all experiments, including baselines, are repeated five times with different sample trainings, seeds, and reported as averages. Additional details are provided in Appendix Section F.

## 4.1 MAIN RESULTS

**Insight 1: Agents can self-improve themselves at test-time even when training on just one sample.** As our main result, we evaluate our test-time self-improvement (TT-SI) on three different agentic benchmarks: NexusRaven, SealTool, and API-Bank,sing both direct zero-shot inference and majority-vote as self-consistency in Table 1. Here we check the effect of TT-SI with SFT by only using one sample generated by **G** and identified through **H**. TT-SI improves the baseline with a 5.36% absolute gain (60.26%→65.62%) in direct inference and 3.52% with majority vote, which shows TT-SI enables agents to self-improve with only one training instance per uncertain case during inference. We also examine a variant of TT-SI as test-time distillation (TT-D), where twhere **G**'s self-generated data is replaced with `GPT5-mini` outputs. TT-D further improves over TT-SI by 2.71% (direct) and 1.17% (majority-vote), indicating that higher-quality training signals provide modest but consistent additional gains.

**Insight 2: TT-SI outperforms inductive SFT with orders of magnitude less data.** We compare TT-SI against in-context learning (ICL, 1-shot) and supervised fine-tuning (SFT) on SealTool, which provides an official training split of ∼13k samples (Figure 2, left). TT-SI with SFT (72.43%) surpasses all three baselines and exceeds standard inductive SFT (70.20%) by 2.23% accuracy. Notably, TT-SI achieves this improvement using only 190 uncertain cases (each paired with one synthetic example) rather than the full 12k training set. This corresponds to roughly **68× fewer samples**, yet delivers better accuracy, highlighting TT-SI as an effective alternative to conventional learning approaches.

**Insight 3: When training is infeasible, Test-Time Self-Improvement with ICL offers a fast alternative.** We extend TT-SI to an in-context learning (ICL) setting (Figure 2, left), where generated examples are inserted directly into the prompt rather than used for fine-tuning. TT-SI with ICL achieves a slight improvement over the base model (66.37%→66.38%) and even outperforms the standard ICL baseline (67.74%) that leverages SealTool's training split. This highlights ICL as a

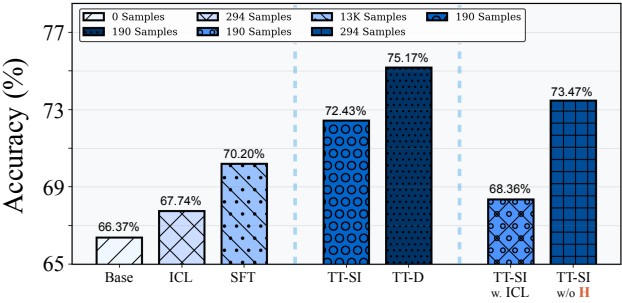 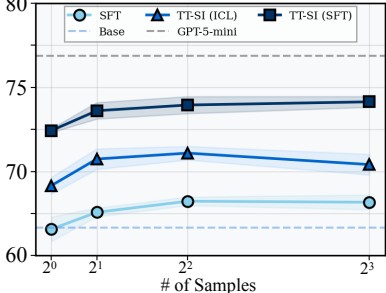

Figure 2: **Experimental Results on SealTool. Left**: Accuracy comparison of TT-SI against standard baselines (left-most) and its variants (middle), including ablations (right-most). **Right**: Scaling behavior under different adaptation strategies with varying numbers of samples. Shaded regions show variance over five runs; dashed lines denote baseline references.

training-free, low-overhead alternative to inductive methods. This improvement likely to be from enhanced model certainty: TT-SI generates demonstrations that boost the model's confidence in the correct output format and reasoning process, increasing the likelihood of accurate predictions without relying on external training data.

**Insight 4: Uncertainty filtering balances accuracy and efficiency.** Because TT-SI operates at inference time, efficiency is critical. In our design, **H** identifies uncertain samples for targeted adaptation, while certain ones are passed directly to the base model. To assess its effect, we also evaluate a variant that treats all test inputs as uncertain (TT-SI w/o **H**) in Figure 2 (left). This achieves only a marginal $+1.04\%$ gain but requires adapting on all 294 test samples rather than 190 samples (an additional 104 LoRA weights to learn) resulting in higher cost. Thus, the slight accuracy gain is outweighed by the efficiency loss, underscoring the importance of uncertainty filtering for practical test-time adaptation. The estimator's accuracy is further detailed in Section D.

## 4.2 ABLATION STUDIES AND ANALYSIS

**Insight 5: Data scaling on OOD data highlights the limits of SFT and the strength of TT-SI.** We examine data scaling for both standard SFT and TT-SI (Figure 2, right). ere, we leverage the state-of-the-art xLAM function-calling dataset (Zhang et al., 2025) for SFT in an out-of-distribution (OOD) setting. For each scale (1, 2, 4, 8), we sample five subsets for training and report averages with standard deviations. TT-SI consistently outperforms SFT across all scales, with improvements growing as more uncertain examples are incorporated, highlighting the importance of uncertainty-guided data and the value of targeted test-time learning. Moreover, the training-free variant of TT-SI with ICL also surpasses standard SFT on a strong dataset using the same data amounts per scale, which shows that even without a dedicated training split or fine-tuning, test-time approaches can outperform SFT under same conditions on OOD data.

**Insight 6: Optimal $\tau$ improves efficiency with minimal accuracy loss.** We investigate the impact of the uncertainty threshold $\tau$ on TT-SI performance and efficiency in Table 2. First, TT-SI improves accuracy regardless of $\tau$, surpassing the base of 66.37% in all cases. A high $\tau$ (approaching 1) selects all samples, yielding the highest accuracy (73.47%) but requiring updates for all 294 instances, resulting in substantial computational overhead for marginal gains. For example, $\tau = 0.95$ achieves 72.43% accuracy with only 190 updates (35% fewer), preserving near-optimal performance. In contrast, a low $\tau = 0.35$ minimizes false positives (FPR=0.09) but misses many errors (TPR=0.42),

| $\tau$ / Setting | TPR | FPR | Unc. ($\Delta\%$) | Acc. |
|---|---|---|---|---|
| Base | – | – | – | 66.37 |
| 0.35 | 0.42 | 0.09 | 51 (17%) | 68.10 |
| 0.95 | 0.96 | 0.53 | 190 (65%) | 72.43 |
| No Unc. (all) | 1.00 | 1.00 | 294 (100%) | 73.47 |

Table 2: **Impact of $\tau$ on TT-SI.** Effect of $\tau$ on uncertain samples, efficiency, and accuracy.

lowering accuracy to 68.10%. Thus, $\tau = 0.95$ offers an effective balance, capturing most errors while avoiding redundant updates and optimizing the cost-performance trade-off, akin to human learning focused on uncertain cases (See Section D for more details on uncertainty calculations).

**Insight 7: TT-SI improves both small and large Qwen models, with larger relative gains for smaller models.** To assess whether TT-SI scales across architectures of different capacities, we conduct experiments with `Qwen2.5-1.5B-Instruct` and its larger counterpart `Qwen2.5-7B-Instruct` (Figure 3). On the smaller model, TT-SI yields a substantial +5.76% absolute gain (66.67→72.43), while on the larger model it delivers a +3.02% gain (80.95→83.97). These improvements indicate that TT-SI consistently enhances performance irrespective of model size, supporting its architectural generality. Interestingly, the relative boost is more pronounced for smaller models, underscoring potential of small agents as an efficiency-oriented strategy for practical deployments (Belcak et al., 2025).

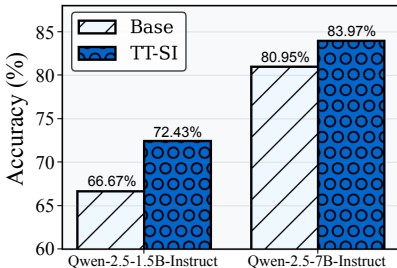

Figure 3: **Model Scaling.** Model Scale and Architectural Generalization with `Qwen2.5-7B-Instruct`.

## 5 DISCUSSIONS

**Conclusion** In this work, we investigate *test-time self-improvement* (TT-SI) for language-based agents that (i) measures uncertainty with **Uncertainty Estimator** (**H**) to decide whether to act directly or adapt, (ii) when uncertain, synthesizes targeted training instances with **Data Synthesis Function** (**G**), and (iii) performs lightweight updates via **Test-Time Fine-tuning** (**T**). We demonstrate that TT-SI improve agents performance during inference, even training with one instance. Across different benchmarks, TT-SI consistently improves test-time performance, while achieving higher accuracy and efficiency than other standard inductive learning baselines. We further analyze the variants of TT-SI, the impact of each component, and key takeaways. Our results reveal the potential of TT-SI, suggesting significant promise for test-time learning in the development of self-evolving agents.

**Impact Statement** Our goal is not to propose a specific uncertainty metric or data generator, but rather a novel algorithm that integrates test-time learning with self-awareness, self-augmentation, and self-improvement for agentic NLP tasks. The design of TT-SI is modular: stronger update rules can replace SFT within **T**, improved uncertainty quantification can plug into **H**, and better data generation can substitute for **G**. We believe that, equipped with a perfect **Uncertainty Estimator** (**H**) that renders the model self-aware of its knowledge and capabilities, a precise **Data Synthesis Function** (**G**) capable of generating diverse yet distributionally aligned samples from uncertain subspaces for self-augmentation, and an effective update mechanism **Test-Time Fine-tuning** (**T**) any scenario becomes learnable in a manner akin to human learning (e.g., a student mastering challenging concepts while preparing for an exam), thereby guiding us toward the realization of a *Master Algorithm*.

**Limitations** While TT-SI demonstrates exciting and promising results, it has limitations. First, identifying uncertain samples requires a threshold $\tau$. Although our ablations show that performance gains are consistent across different values of $\tau$ (Section 4.2), the best performance is sensitive to this choice. Principled methods for learning this threshold autonomously in uncertainty calibration domain remain an open challenge (Bakman et al., 2025). Finally, TT-SI is inherently bounded by the capacity of the base model parameters $\theta$. If the knowledge required to solve a task is absent from the pretrained model (e.g., a newly introduced medical concept), self-improvement alone cannot recover it; in such cases, external knowledge integration through retrieval or search mechanisms is necessary, pointing towards self-evolving agents.

**Future Work** Beyond self-improvement, a key direction is enabling *self-evolving* agents capable of continual and transferable learning (Gao et al., 2025). In TT-SI, temporary and sample-specific LoRA updates are applied and discarded after inference. An important next step is investigating whether improvements from one uncertain sample can transfer to others, and how agents can self-assess when such transfer is possible. Another promising direction is adaptive data generation, where the model itself determines how many synthetic examples are needed for a given uncertain case rather than relying on fixed hyperparameters (Zweiger et al., 2025). Furthermore, our current framework optimizes only the agent, not the data generator; a co-evolutionary setup like dual-learning, where both the agent and generator adapt to each other (e.g., mutual learning), could further enhance performance (Zhou et al., 2025). Finally, extending TT-SI to domains such as mathematics or medicine presents an opportunity to explore how domain-specific uncertainty and knowledge structures interact with self-improvement (Zhao et al., 2025).

ETHICS STATEMENT

This research investigates test-time self-improvement (TT-SI) for language-based agents using publicly available benchmarks. No personally identifiable information or sensitive human data was collected or used. All experiments rely on synthetic datasets, open-source corpora, or established evaluation benchmarks. We recognize potential risks, such as biases from LLM-based data generation and training, which are mitigated by the nature of test-time training (TTT), as we do not store any model weights. We further mitigate these by (i) reporting transparent experimental details, (ii) restricting scope to controlled research settings, and (iii) emphasizing reproducibility and responsible use. Our research adheres to the ICLR Code of Ethics by prioritizing fairness, privacy, and scientific integrity.

REPRODUCIBILITY STATEMENT

We truly believe transparency and reproducibility are essential for future and successful research. We provide our novel algorithm in Algorithm 1. Since our method often operates with very limited data (e.g., a single sample trainings), relatively high variance can be expected. To address this, all experiments (including baselines) are repeated five times with different random seeds, and we report averaged results in Section 4. Where appropriate (e.g., in line plots), we also present standard deviations. All experiments are conducted on a single NVIDIA A40 GPU. For evaluation, we use publicly available and widely used benchmarks (e.g., API-Bank (Li et al., 2023), SealTool (Wu et al., 2024), NexusRaven (Srinivasan et al., 2023)). Additional implementation details, including hyperparameters and evaluation metrics, are provided in Section F. Together, these resources ensure transparency and allow independent reproduction of all reported results.

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

APPENDIX

## A  TERMINOLOGY: SELF-IMPROVING AND SELF-EVOLVING AGENTS

In the context of agentic systems powered by LLMs, distinguishing between *self-improving* and *self-evolving* agents is crucial for understanding their capabilities, limitations, and paths toward advanced intelligence. This appendix provides definitions, key differences, and technical insights drawn from recent literature.

- **Self-Improving Agents:** Self-improving agents refer to systems that autonomously enhance their own performance on specific tasks through iterative self-refinement mechanisms, without requiring any external intervention. These agents typically leverage their own outputs to identify weaknesses and generate improvements, often focusing on a fixed architecture or model. For instance, an agent might use self-reflection to critique its reasoning on a task, then refine its prompts or generate synthetic data for fine-tuning. The scope is generally task-specific, emphasizing efficiency gains within bounded domains, such as data analysis or coding, without altering the underlying system structure.

- **Self-Evolving Agents:** Self-evolving agents are designed for broader, continuous adaptation across dynamic environments and sequential tasks, enabling lifelong learning and generalization. These agents evolve not only parametric components (e.g., model weights) but also non-parametric elements like memory, tools, prompts, and architecture. This allows agents to handle open-ended scenarios, such as real-world feedback loops in interactive environments.

Overall, self-improving agents target specific task optimization via iterative refinement, while self-evolving agents emphasize holistic system evolution for adaptability to novel environments and long-term generalization.

## B  OTHER EXAMPLES FROM SELF-REGULATED LEARNING

**Student Homework.**  Our paradigm of test-time self-improvement (TT-SI) also draws inspiration from how students engage in self-learning (Zimmerman, 2002). When faced with uncertainty, such as being unsure how to solve a homework problem (*self-awareness*), students often seek out related examples from textbooks and online resources (*self-augmentation*) to resolve their knowledge gap and build confidence in solving similar tasks (Winne & Hadwin, 1998) (*self-improvement*). This process is closely aligned with theories of metacognition and self-regulated learning, where learners actively identify their knowledge gaps and pursue targeted resources to close them (Nelson, 1990; Winne & Hadwin, 1998; Zimmerman, 2002). Unlike classical active learning in machine learning (Settles, 2009), which deliberately queries an oracle or annotator for novel and informative examples, our method generates additional data automatically without external supervision. Moreover, while student learning often benefits from diverse perspectives and human explanations, our approach focuses on generating semantically similar but slightly varied problem instances to refine the model's performance. This analogy highlights the natural intuition behind TT-SI while underscoring its distinct contribution as an automated, uncertainty-driven, and cost-efficient alternative to data collection.

**Sport Analogy.**  Lets also consider a running back (RB) in American football honing skills for the NFL Combine, whose performance relies heavily on lower-body strength and the rate of force development for explosive acceleration. If an athlete's primary weakness lies in short-burst speed and rapid change-of-direction, a targeted regimen emphasizing plyometric drills, resisted sprints, and eccentric–concentric coupling work can directly address this limitation. In contrast, allocating significant training time to non-specific full-body hypertrophy (e.g., frequent bench pressing or isolated arm work) not only increases recovery demands and neuromuscular fatigue but can also lead to excess non-functional muscle mass, which may reduce stride frequency and overall sprint velocity. By diagnosing the limiting factor (*self-awareness*), incorporating performance-specific drills (*self-augmentation*), and progressively refining execution through repeated exposure (*self-improvement*), the athlete can achieve more meaningful outputs without the performance trade-offs of untargeted training.

## C PREVIOUS WORK ON LARGE LANGUAGE MODELS (LLMS) AND AGENT FINE-TUNING

The de-facto approach to equip LLMs with new capabilities is to collect task-specific corpora and fine-tune on them (Kumar et al., 2025), with such datasets either curated through human annotation (Ouyang et al., 2022) or synthesized by LLMs (Wang et al., 2023). Following these advancements, LLM-based agents have emerged (Yao et al., 2023), where models interact with external tools and APIs rather than producing text alone, which require learning tool-use skills and handling structured inputs and outputs (Patil et al., 2024). Training LLMs for such agentic skills has led to the exploration of effective dataset design and tuning strategies aimed at improving generalizability (Zeng et al., 2024; Mitra et al., 2024; Chen et al., 2024). However, this inductive approach is prone to catastrophic forgetting when transferred across different environments, requires costly data generation pipelines, and does not guarantee consistent gains over strong base models. In contrast, to the best of our knowledge, our work introduces the first application of TTT to LLM-based agents by enabling temporary parameter updates during inference and therefore avoids catastrophic forgetting and reduces dependence on large offline datasets. Furthermore, considering training efficiency, we incorporate selective data usage by using only the most informative samples for the model, rather than redundantly training on already well-understood instances.

## D UNCERTAINTY ESTIMATION RESULTS

We show the performance of **Uncertainty Estimator** (**H**) with `Qwen-2.5-1.5B-Instruct` on Nexus-Raven (Srinivasan et al., 2023) in Figure 4. For every test input, we (i) obtain RSS confidence scores $p_n$ via Equation (3), (ii) compute the softmax-difference $u(x_i) = p^{(1)} - p^{(2)}$, and (iii) mark the prediction as *uncertain* (i.e., select it for adaptation) when $u(x_i) < \tau$. We study how the choice of threshold $\tau$ affects performance; Figure 1 (left) summarizes the main findings. As the threshold $\tau$ for the softmax-difference $u(x_i)$ increases ($0.35 \rightarrow 0.99$), the condition $u(x_i) < \tau$ for being uncertain becomes less stringent, leading to more samples being identified as uncertain and routed for downstream adaptation. Raising $\tau$ monotonically increases both the true positive rate (TPR, i.e., correctly flagging the model's wrong predictions as uncertain) and the false positive rate (FPR, i.e., incorrectly flagging the model's correct predictions as uncertain). Ideally, the goal is to **maximize TPR** while **keeping FPR as low as possible**. When we increase $\tau$, TPR rises steadily from 16% at $\tau = 0.35$ to 88% at $\tau = 0.95$ and FPR remains extremely low across all thresholds, only increasing from 1% to 17% as $\tau$ approaches 0.95 and 0.99. The overall discrimination ability of the estimator, measured by Youden's J statistic (Ruopp et al., 2008) measured by TPR $-$ FPR and peaks at 71.0% when $\tau$ is set to 0.95.

At this optimal threshold, the estimator captures nearly 88% of model errors as uncertain, while only miss classifying 17% of correct answers. This reflects a strong balance between *sensitivity* and *specificity*, demonstrating the effectiveness of our (**H**). For all experiments, we adopt $\tau = 0.95$ as the default threshold, as it consistently achieves the best trade-off between identifying the majority of erroneous outputs and minimizing unnecessary intervention on correct predictions. However, $\tau$ is an hyperparameter and by tuning $\tau$, one can flexibly adjust the stringency of uncertainty filtering to match the requirements of specific downstream tasks or adaptation budgets, ensuring both effective error coverage and efficient resource allocation. Further details and results for the **Uncertainty Estimator** (**H**) are provided in Section D. Future work should establish more effective methods to automatically determine the optimal $\tau$, as this remains a central challenge in the domain of uncertainty estimation.

## E DATA GENERATION DETAILS

The implementation of **Data Synthesis Function** (**G**), which is triggered for each uncertain sample $x_i$, employs the agent itself for data synthesis ($\mathcal{L}_{\text{gen}}$) as *self-augmentation*. For each generation instance, $\mathcal{L}_{\text{gen}}$ is provided with a carefully hand-crafted prompt $\mathcal{P}$ (See Figure 6), the uncertain input $x_i$ serving as the direct seed (critically, without its corresponding label $y_i$), and a specified number of samples $K$ to generate. The model then produces $K$ new input-output pairs, denoted as $\{(x'_{ij}, y'_{ij})\}_{j=1}^{K}$. This seed-based generation process, inspired by self-instruction methodologies (Wang et al., 2023),

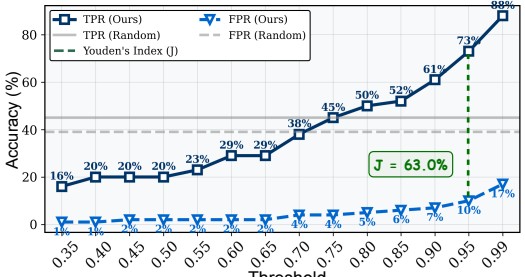

Figure 4: **Uncertainty Estimator Performance with Qwen-2.5-1.5B-Instruct on Nexus-Raven.**
**Left:** Effect of varying the softmax-difference threshold $\tau$ on true positive rate (TPR) and false positive rate (FPR). Increasing $\tau$ broadens the definition of "uncertain," steadily raising TPR (i.e., correct detection of erroneous predictions) from 16% to 88% while keeping FPR (i.e., mislabeling correct predictions) below 17%. Youden's index ($J = \text{TPR} - \text{FPR}$) peaks at $\tau = 0.95$, balancing sensitivity and specificity, and is adopted as the default threshold in subsequent experiments. **Right:** Comparison of uncertainty estimation methods. Our estimator (**H**) in Section 3.1 outperforms baselines (Random, Trivial, Perplexity) across all metrics, achieving the highest F1 (61.19) and Youden's J (63.29).

guides $\mathcal{L}_{\text{gen}}$ to produce variants that maintain the core semantic meaning and task relevance of $x_i$ while introducing controlled surface-level variations. By *synthesizing data in this on-the-fly* manner for each uncertain instance, we facilitate targeted and timely model adaptation, aiming to improve performance on precisely the types of queries the model struggles with, as they are encountered.

### E.1 EXPERIMENTAL RESULTS

For each uncertain input $x_i$ detected by the procedure in **H** we synthesize exactly one new example ($K = 1$) using the same LLM (i.e., `Qwen-2.5-1.5B-Instruct`). The generator receives only the instruction and query of $x_i$—never the gold label—and produces both a revised input and its answer, thereby creating a temporary, query-specific dataset $\mathcal{D}_i$ that is used immediately for inference-time adaptation.

Interpreting and understanding how our **Data Synthesis Function** (**G**) operates is essential for understanding the effectiveness of our generations. To this end, we embed (Reimers & Gurevych, 2019) all SealTool test samples, an uncertain example $x_i$ from this set, and ten self-generated queries for $x_i$ produced by `Qwen-2.5-1.5B-Instruct` into a two-dimensional semantic space using UMAP (McInnes et al., 2018). As visualized in Figure 5, the generated samples form a compact cluster in the embedding space, closely aligned with both in each other and the corresponding uncertain input. This spatial proximity suggests that our data synthesis function **G** can yields mutually consistent and semantically faithful examples, effectively bridging the gap for adaptation to challenging queries.

**Detailed Qualitative Analysis of Synthetic Query Generation.** To provide a more comprehensive qualitative evaluation of the quality and diversity of our self-generated synthetic queries, focus on the semantic embedding space derived from the SealTool dataset (Wu et al., 2024). We begin by encoding textual data into dense

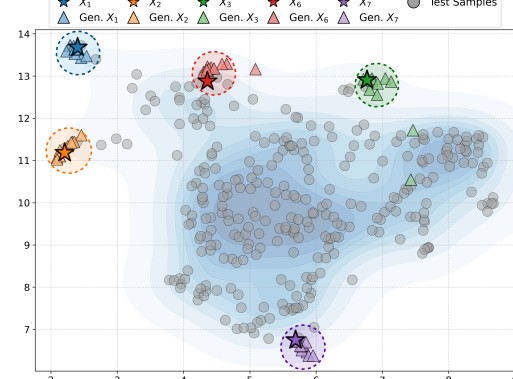

Figure 5: **Self-Generated Data Visualization.** All test samples (circles) are projected into a two-dimensional semantic space via UMAP, and shown with the density contour distributions. The star denotes the uncertain input $x_i$, and triangles indicate 10 randomly sampled, self-generated synthetic queries from uncertain sample $x_i$. Generated samples are tightly clustered and situated near $x_i$, demonstrating distributional alignment of **G**.

vectors. Each sample is represented as a concatenation of the system prompt, user query, and output response (or equivalent instruction-input-output triples for generated data). These are embedded using the Sentence-BERT model with `all-mpnet-base-v2` (Reimers & Gurevych, 2019), which produces 768-dimensional vectors optimized for semantic similarity in natural language tasks. The high-dimensional embeddings are then projected into a two-dimensional latent space using Uniform Manifold Approximation and Projection (UMAP) (McInnes et al., 2018), a nonlinear dimensionality reduction algorithm that preserves both local and global topological structures more effectively than alternatives like t-SNE (van der Maaten & Hinton, 2008). We configure UMAP with 15 neighbors to balance local clustering and global layout, and set minimum distance as 0.1 to allow moderate spread in low-density regions, facilitating the identification of outliers such as uncertain samples.

---

**Data Generation Prompt**

You are given an instruction, input, and example sample as seed, but not labeled output. You must generate new synthetic examples that closely match the original uncertain scenario.

1. Create distinct variants of the seed by altering names, context, or wording but no variant may duplicate the original.
2. Your response format must be: { "instruction": "<instruction>", "input": "<input>", "output": "<output>" }
3. The "output" field must be a function calling JSON object with the following structure: [{"name": "Tool name", "parameters": {"Parameter name": "Value",...}},..]

### Number of Examples
Generate <number> examples.

### Seed Example
<seed_example>

### Generated Examples
Your Response:

---

Figure 6: Data Generation Prompt for uncertain samples with LLM.

## F   IMPLEMENTATION DETAILS OF TT-SI

We firmly believe that transparency and detailed reporting are essential for both understand the approach in-depth and advancing future research. Accordingly, we do our best to provide complete descriptions of each component of the proposed TT-SI framework: **Uncertainty Estimator** (**H**) (Section 3.1), **Data Synthesis Function** (**G**) (Section 3.2), and **Test-Time Fine-tuning** (**T**) (Section 3.3). In all steps, we use `Qwen2.5-1.5B-Instruct`[1] from HuggingFace checkpoints, running on a single NVIDIA A40 GPU.

### F.1   UNCERTAINTY ESTIMATION

For implementing **H**, we use the HuggingFace Transformers library (Wolf et al., 2019), as it provides straightforward access to token logits and confidence estimates through the `AutoModelForCausalLM` class, unlike vLLM. We do not apply temperature scaling at this step. To estimate uncertainty, we directly input the test sample instruction as a query and merge all available function names extracted via regex operations. The confidence scores for each candidate function are computed using Equation (2) and normalized with RSS as formulated in Equation (3). On SealTool, labeling a sample as uncertain requires on average 0.88 seconds.

### F.2   DATA GENERATION

Once an uncertain sample is identified with **H**, we generate $K$ similar samples using **G**. This is done with the prompt shown in Figure 6, where the model is asked to create slight variations of the sample (but not the exact same sample) along with corresponding labels. The uncertain sample is inserted

---

[1]https://huggingface.co/Qwen/Qwen2.5-1.5B-Instruct

into the prompt as a seed (<seed>) (Wang et al., 2023), and $K$ is set as a hyperparameter (<number>) by replacing special tokens. We then extract the generated samples. We study two variants: TT-SI and TT-D. In TT-SI, the same `Qwen2.5-1.5B-Instruct` model generates its own synthetic samples, while in TT-D, sample generation is performed by `GPT-5-mini`[2]. For TT-SI, we use vLLM with the following settings: temperature 0.7, repetition penalty 1.1, top-$k = 20$, and top-$p = 0.8$, consistent with the recommended `Qwen2.5-1.5B-Instruct` generation configuration, and maximum length set to 32768. Because of the model's small scale, sometimes parsing errors occur, where the model may omit some JSON strings. We allow up to 5 retries; in practice, errors are usually resolved within the second or third attempt. For `GPT-5-mini`, we use the standard OpenAI API without temperature adjustments or additional decoding strategies. The computational cost of using GPT-5-mini is negligible, effectively zero for a single experiment. On average, generating one sample with TT-SI takes approximately 3.55 seconds. A qualitative analysis of the data generation process is provided in Section E.

### F.3 TRAINING

After obtaining the $K$ generated samples from **G**, we directly perform test-time fine-tuning with **T**. For training, we use LLaMA-Factory (Zheng et al., 2024), chosen for its optimized implementation and user-friendly CLI. All fine-tuning is conducted with Parameter-Efficient Fine-Tuning (PEFT) via LoRA (Hu et al., 2022). We set the LoRA parameters to $rank = 8$ and $\alpha = 16$, applied to all linear layers. Training runs for 3 epochs with a fixed learning rate of $1.0 \times 10^{-4}$, a warm-up ratio of 0.03, and batch size of 1. Despite the short training, we use a cosine scheduler by default. Otherwise, we keep the default configuration parameters of LLaMA-Factory and HuggingFace without further modifications. For inference on the fine-tuned models, we adopt the same vLLM decoding settings as in data generation: temperature 0.7, repetition penalty 1.1, top-$k = 20$, and top-$p = 0.8$. All trainings follow the Alpaca-style data format (Taori et al., 2023), where the instruction and input fields are zero-padded, and the loss is computed only on the output field. Training a single sample with LLaMA-Factory takes on average 4.55 seconds. On the other hand, our reported timings exclude I/O operation overheads from checkpoint loading, merging, and saving, as these are highly implementation-dependent, which is discussed in Section G.

### F.4 EVALUATION

We evaluate our method on three established agent benchmarks: NexusRaven (Srinivasan et al., 2023), SealTool (Wu et al., 2024), and API-Bank (Li et al., 2023). **NexusRaven** focuses on realistic software operation tasks, particularly in domains such as cybersecurity and enterprise applications. It is designed to test high-fidelity function execution in business scenarios, featuring long and diverse tool invocations across 65 distinct APIs with a total of 318 samples (see Figure 7 for an example). **SealTool** is one of the most extensive and recent benchmarks, comprising 4,076 APIs spanning diverse domains. Its latest version is designed to minimize potential data leakage, making it a robust benchmark for tool-use evaluation. In our experiments, we use the curated test set of 294 samples (see Figure 8 for an example). **API-Bank** contains 314 multi-turn conversations with 753 distinct API calls. It evaluates an LLM's ability to select appropriate functions and arguments in realistic dialogue settings. Following prior work, we focus on 316 samples from Levels 1 and 2, which balance task complexity and data availability (see Figure 9 for an example). Across all benchmarks, we assess whether the model produces the correct function name, arguments, and corresponding values/types. A prediction is considered correct only if *all* components match the ground truth simultaneously; otherwise, it is marked as incorrect.

## G ADDITIONAL RUN-TIME OVERHEAD AND RESOURCE USAGE ANALYSIS

We examine run-time of each steps of TT-SI on SealTool dataset against standard SFT under identical conditions. We use HuggingFace (Wolf et al., 2019) for confidence estimation with **H**, vLLM (Kwon et al., 2023) for data synthesis with **G** and inference, and LLaMA-Factory (Zheng et al., 2024) for trainings with **T**. On per-sample average, **H** requires $0.88s$ to assess uncertainty; for uncertain samples, **G** generates synthetic variants in $3.55s$, followed by **T** training in $4.55s$, and inference in

---

[2]https://platform.openai.com/docs/models/gpt-5-mini

$1.26s$. This totals $10.24s$ for uncertain samples and $2.14s$ otherwise, amounting to $2,168.2s$ ($\sim 36$ minutes) for 190 updates. In contrast, SFT requires $7,966.6s$ ($\sim$2h12m) to train on SealTool's 13K-sample split. Despite training on $\sim 68\times$ fewer samples, TT-SI delivers a $3.7\times$ wall-clock speed-up. The majority of additional time is due to model merging and file-saving operations used for weight operation after training. HuggingFace provides an option to directly use merged weights after training, thus we exclude I/O operations from clock-time calculations.

## H  USE OF LLMs

In this work, LLMs were used in this work for three purposes: (i) as base models under study for test-time training, (ii) as baselines for empirical comparison, and (iii) for minor assistance in refining the readability of this manuscript. Both open-source (e.g., Qwen (Yang et al., 2025)) and closed-source models (e.g., GPT-5-mini[3]) were employed for training and data-generation. The prompt used for improving writing quality was similar to *"Please make more clear sentence, making sure to remove any grammatical mistakes."* Importantly, all scientific ideas, methods, experiments, and conclusions originate from the authors. When LLMs were used for language refinement, outputs were carefully reviewed to prevent the introduction of hallucinated or incorrect content, ensuring that all arguments, findings, and perspectives are solely those of the authors.

---

**NexusRaven Test Sample Example (ID: 317)**

You are an advanced assistant capable of using tools to help the user. You may call one or more functions to assist with the user query. For any user request that requires a function, respond by returning a function call inside **<tool_call>**...**</tool_call>** XML tags, with a JSON object specifying the "name" of the function and the "arguments".

**Task Instruction**
In order to complete the user's request, you need to select one or more appropriate tools from the following tools and fill in the correct values for the tool parameters. Your specific tasks are:
1. Make one or more function/tool calls to meet the request based on the question.
2. If none of the functions can be used, point it out as an empty list and refuse to answer.
3. If the given question lacks the parameters required by the function, also point it out.

**Output Format**
For each function call, return a JSON object with function name and arguments within **<tool_call></tool_call>** XML tags:
**<tool_call>**[ {"name": "<function-name>", "arguments": {"arg1": "value1", "arg2": "value2", ...} , ...] **</tool_call>**
If no function call is needed, please directly output an empty list '[]' as **<tool_call>[]</tool_call>**.

**Available Tools**:
In your response, you can use the following tools:
**<tools>**
1. Name: verifyUSAddress
Description: Verify a given US address to ensure it meets USPS standards and is deliverable.
Parameters: {'addressLine1': {'type': 'str', 'description': 'The primary address line, including street number and name.', 'required': True}, 'addressLine2': {'type': 'str', 'description': 'The secondary address line, such as apartment or suite number.', 'required': True}, 'city': {'type': 'str', 'description': 'The city of the address.', 'required': True}, 'state': {'type': 'str', 'description': 'The state or territory of the address.', 'required': True}, 'zipCode': {'type': 'str', 'description': 'The 5-digit ZIP code of the address.', 'required': True}}
2. Name: standardizeUSAddress
Description: Standardize a given US address to create consistency and accuracy in addressing.
Parameters: {'addressLine1': {'type': 'str', 'description': 'The primary address line, including street number and name.', 'required': True}, 'addressLine2': {'type': 'str', 'description': 'The secondary address line, such as apartment or suite number.', 'required': True}, 'city': {'type': 'str', 'description': 'The city of the address.', 'required': True}, 'state': {'type': 'str', 'description': 'The state or territory of the address.', 'required': True}, 'zipCode': {'type': 'str', 'description': 'The 5-digit ZIP code of the address.', 'required': True}}
**</tools>**

**Question**
User: I'm organizing a mailing list for my business, and I want to make sure all the addresses are standardized. Can you help me standardize this address? 456 Street, Suite 7891, Los Angeles, CA, 90011.

**Your Response**: **<tool_call>**[ {"name": "standardizeUSAddress", "arguments": {"addressLine1": "456 Street", "addressLine2": "Suite 7891", "city": "Los Angeles", "state": "CA", "zipCode": "90011"} } ] **</tool_call>**

Figure 7: Sample example from NexusRaven test data.

---

[3]https://platform.openai.com/docs/models/gpt-5-mini

**SealTool Test Sample Example (ID: 4)**

You are an advanced assistant capable of using tools to help the user. You are given a conversation between a user and an assistant, together with the available tools.
You may call one or more functions to assist with the user query.
You will be provided with a set of Available Functions inside **<tools>...</tools>** tags.
For any user request that requires a function, respond by returning a function call inside **<tool_call>...</tool_call>** XML tags, with a JSON object specifying the "name" of the function and the "arguments".

**Task**
1. Think and recall relevant context, analyze the current user goal.
2. Refer to the previous dialogue records in the conversations, including the user's queries.
3. Decide on which tool to use from **Available Tools** and specify the tool name.
4. At the end, you need to output the JSON object of the function call inside the **<tool_call>** and **</tool_call>** tags.
5. Output format of the function calls must be EXACTLY like in the **Output Format** section, the function calls must be a list of JSON objects, each object must have a "name" key and an "arguments" key.
6. This year is 2023.

**Output Format**
For each function call, return a JSON object with function name and arguments within **<tool_call></tool_call>** XML tags:
**<tool_call>**[ {"name": "<function-name>", "arguments": {"arg1": "value1", "arg2": "value2", ...} , ...] **</tool_call>**

**Available Tools**
**<tools>**
1. Name: analyzeSample
Description: Analyze a given sample using analytical chemistry techniques
Field: Chemistry/Analytical chemistry
Parameters: {'sample': {'type': 'str', 'description': 'The sample to be analyzed'}, 'method': {'type': 'str', 'description': 'The analytical method to be used for analysis (e.g., chromatography, spectroscopy)'}, 'instrument': {'type': 'str', 'description': 'The instrument or equipment to be used for analysis (e.g., gas chromatograph, mass spectrometer)'}, 'conditions': {'type': 'str', 'description': 'Any specific conditions required for the analysis (e.g., temperature, pressure)'}}
Required: [sample, method]
Responses: {'results': {'type': 'str', 'description': 'The analysis results containing information about the sample'}}
2. Name: analyzeEvidence
Description: Analyze the chemical evidence collected from a crime scene
Field: Chemical Engineering/Forensic engineering
Parameters: {'evidence_type': {'type': 'str', 'description': 'The type of evidence to be analyzed (e.g., DNA, fingerprints, blood, fibers)'}, 'method': {'type': 'str', 'description': 'The method or technique to be used for analysis (e.g., spectroscopy, chromatography, microscopy)'}, 'sample': {'type': 'str', 'description': 'The sample or specimen to be analyzed (e.g., crime scene swab, hair strand, fabric sample)'}}
Required: [evidence_type, method, sample]
Responses: {'analysis_results': {'type': 'str', 'description': 'The results of the chemical analysis of the evidence'}, 'conclusion': {'type': 'str', 'description': 'The conclusion drawn from the analysis'}}
3. Name: getSampleSize
Description: Retrieve the sample size of a mixed methods research study
Field: Research/Mixed Methods Research
Parameters: {'study_id': {'type': 'str', 'description': 'The unique identifier of the research study'}}
Required: [study_id]
Responses: {'sample_size': {'type': 'int', 'description': 'The sample size of the research study'}}
4. Name: getFabricComposition
Description: Retrieve fabric composition information for a specific clothing item
Field: Fashion/Fashion Technology
Parameters: {'clothing_item': {'type': 'str', 'description': 'The type of clothing item for which you want fabric composition (e.g., t-shirt, jeans, dress)'}, 'brand': {'type': 'str', 'description': 'The brand of the clothing item (e.g., Nike, Zara, Gucci)'}}
Required: [clothing_item]
Responses: {'composition': {'type': 'str', 'description': 'The fabric composition of the specified clothing item'}, 'brand': {'type': 'str', 'description': 'The brand of the clothing item'}}
5. Name: evaluateDataBias
Description: Evaluate data bias in a dataset
Field: Data Analysis/Data Ethics
Parameters: {'dataset': {'type': 'str', 'description': 'The dataset to evaluate for bias (e.g., hiring records, loan applications)'}, 'protected_attributes': {'type': 'str', 'description': 'The protected attributes to consider for bias assessment (e.g., gender, race)'}, 'measures': {'type': 'str', 'description': 'The bias assessment measures to be used (e.g., disparate impact, statistical parity index)'}, 'reference_group': {'type': 'str', 'description': 'The reference group to compare with for bias assessment'}}
Required: [dataset, protected_attributes]
Responses: {'bias_score': {'type': 'float', 'description': 'The overall bias score of the dataset'}, 'protected_attributes_bias': {'type': 'str', 'description': 'Detailed bias assessment for each protected attribute'}}
**</tools>**

**Input**
User: Provide the statistics for the Real Madrid team.

**Your Response**: **<tool_call>**[ {"name": "getTeamStats", "arguments": {"team": "Real Madrid"} } ] textbf**</tool_call>**

Figure 8: Sample example from SealTool test data.

---

**API-Bank Test Sample Example (ID: 0)**

You are an advanced assistant capable of using tools to help the user. You are given a conversation between a user and an assistant, together with the available tools.
You may call one or more functions to assist with the user query.
For any user request that requires a function, respond by returning a function call inside
textbftexttt<tool_call>...</tool_call> XML tags, with a JSON object specifying the "name" of the function and the "arguments".

**Task**
1. Think and recall relevant context, analyze the current user goal.
2. Refer to the previous dialogue records in the conversations, including the user's queries.
3. Decide on which tool to use from **Available Tools** and specify the tool name.
4. At the end, you need to output the JSON object of the function call inside the **<tool_call>** and **</tool_call>** tags.
5. Output format of the function calls must be EXACTLY like in the **Output Format** section, the function calls must be a list of JSON objects, each object must have a "name" key and an "arguments" key.
6. This year is 2023.

**Output Format**
For each function call, return a JSON object with function name and arguments within **<tool_call></tool_call>** XML tags:
**<tool_call>**[ {"name": "<function-name>", "arguments": {"arg1": "value1", "arg2": "value2", ...} , ...] **</tool_call>**

**Available Tools**:
In your response, you can use the following tools:
**<tools>**
1. Name: QueryHealthData
Description: This API queries the recorded health data in database of a given user and time span.
Parameters: {'user_id': {'type': 'str', 'description': 'The user id of the given user. Cases are ignored.'}, 'start_time': {'type': 'str', 'description': 'The start time of the time span. Format: %Y-%m-%d %H:%M:%S'}, 'end_time': {'type': 'str', 'description': 'The end time of the time span. Format: %Y-%m-%d %H:%M:%S'}}
2. Name: CancelRegistration
Description: This API cancels the registration of a patient given appointment ID.
Parameters: {'appointment_id': {'type': 'str', 'description': 'The ID of appointment.'}}
3. Name: ModifyRegistration
Description: This API modifies the registration of a patient given appointment ID.
Parameters: {'appointment_id': {'type': 'str', 'description': 'The ID of appointment.'}, 'new_appointment_date': {'type': 'str', 'description': 'The new appointment date. Format: %Y-%m-%d.'}, 'new_appointment_doctor': {'type': 'str', 'description': 'The new appointment doctor.'}}
**</tools>**

**Conversation**
User: Can you please modify my appointment scheduled for March 25th with Dr. Kim to March 26th with Dr. Lee?
Assistant: Sure, I can help you with that. Please provide me with the appointment ID and the new appointment date and doctor's name.
User: The appointment ID is 34567890 and the new date is March 26th with Dr. Lee.
Assistant: Alright. I'll modify your appointment now.
User: Based on our conversation above, please only make one tool call to solve my need.

**Output**: **[<tool_call>**[{"name": "ModifyRegistration", "arguments": {"appointment_id": "34567890", "new_appointment_date": "2023-03-26", "new_appointment_doctor": "Dr. Lee"}}]**</tool_call>]**

Figure 9: Sample example from API-Bank test data.

