# OpenReview forum: "Self-Improving LLM Agents at Test-Time"
_ICLR.cc/2026/Conference — ICLR 2026 Conference Withdrawn Submission_

### Official Review · Reviewer_CMNh · 2025-10-20

**Soundness:** 4
**Presentation:** 4
**Contribution:** 2
**Rating:** 4
**Confidence:** 4

**Summary:**

This paper proposes a new method named TT-SI which can distill model's own knowledge during inference. It will use an Uncertainty Estimator to estimate whether the model's response is confident or not. If the model's response is not confident, it will generate some similar questions and answers and finetune the model with these data, and then inference again. (If the generation model is not the same model as the original model, its method is named TT-D). The experiments show that the model can improve its performance on the benchmark tasks.

**Strengths:**

1. The experiment is enough and almost give the information that I want to know.

2. The structure of the paper is clear, especially the experiment part is well-designed.

3. It designs a new method in test-time training.

4. The findings is quite interesting.

**Weaknesses:**

1. The code of such test-time training is too high, since once the model is finetuned, the training process will not take any more money, but in this method, it will finetune the model again and again during inference. (It may also waste a lot of generated data since it will reset the papameter). If the benchmarks expand 100x, the SFT's cost will not change but this method will expand 100x. This makes the method not practical.

2. Although I agree this method has positive results, I'm still suspicious about the motivation, even other papers in line 76-87:

As we consider the model "improve", we mostly consider the Pass@1 performance; however, this paper's upper level bound is Pass@1=Major@infinity, which is minor than the Pass@infinity (likely to be 1). And I believe Major@infinity is not a good upper level bound (especially for the hard questions). The improvement may come from the pretrain model's SFT stage or annealing stage, which increases the Major@infinity by training similar data.
As a result, the upper bound may harm the generalizability of the method.

I appreciate the author's hard work, however, because of the weaknesses, I'll give a borderline score to this paper, and I may raise my score if the author can address the concerns in the rebuttal.

**Questions:**

1. Solve the concern in weakness 2.

---

### Official Review · Reviewer_R7kG · 2025-10-25

**Soundness:** 2
**Presentation:** 2
**Contribution:** 2
**Rating:** 2
**Confidence:** 4

**Summary:**

This paper introduces a method called Test-Time Self-Improvement (TT-SI), enabling an LLM agent to refine itself during inference. The core pipeline is:

1. Uncertainty estimation : flag test samples on which the model is “uncertain”;
2. Data synthesis : automatically generate synthetic training data from these samples;
3. Test-time tuning : perform a lightweight LoRA fine-tune on the model with the synthetic data, then infer on the same sample.

Experiments show that the proposed method improves the model’s performance on the test set.

**Strengths:**

1. The paper validates the method’s effectiveness on three test sets.
2. The proposed TT-D variant shows how synthetic-data quality influences performance.
3. The pipeline cleanly modularizes into uncertainty estimation, data synthesis, and test-time fine-tuning, so it is readily extensible.

**Weaknesses:**

1. The core of TT-SI is to generate *k* synthetic training examples that resemble the model-uncertain test point, fine-tune on them, and then re-predict the very same input. This means every uncertain test sample is—directly or indirectly—present in the training signal.
2. Despite the authors' characterization of the method as "test-time adaptation," its central conclusion—that generating training data resembling the test sample and fine-tuning on it boosts performance—is mere common sense and offers no novelty.

**Questions:**

1. The core of TT-SI is to generate *k* synthetic training examples that resemble the model-uncertain test point, fine-tune on them, and then re-predict the very same input. This means every uncertain test sample is—directly or indirectly—present in the training signal. Could the authors clarify how TT-SI is fundamentally different from ordinary “test-set leakage”?
2. Even if TT-SI is not identical to classic leakage, performing one parameter update per uncertain question is impractical.
3. Can the authors provide evidence that TT-SI yields better generalization than fine-tuning on the *full* training set, rather than merely over-fitting to the single “uncertain” example?

---

### Official Review · Reviewer_pfG1 · 2025-10-31

**Soundness:** 1
**Presentation:** 2
**Contribution:** 3
**Rating:** 2
**Confidence:** 3

**Summary:**

The paper introduces TT-SI and TT-D, test-time training approaches for LLM agents. The methods consist of three modules: An uncertainty estimator H to identify test samples used for training, a generation function G to synthesize new examples for training, and a learner T to update the model's parameters with using the synthesized examples. The proposed methods outperform the base model, in-context learning, and supervised fine tuning (SFT) on three agent benchmarks.

**Strengths:**

- The experiments suggest that TT-SI improves results substantially on the tested data sets.
- The result that generated output can be used for model inprovement using TTT is interesting.
- The structuring of the results section into 7 main insights is effective.
- TT-SI is modular: One can mix and match different uncertainty estimators, data synthesis functions, and fine-tuning algorithms.

**Weaknesses:**

- The required inference time increases from 1.26s per example without TT-SI to 10.24s for uncertain examples with TT-SI, or 2.14s for certain examples with TT-SI (Appendix G).
- The largest complaint I have is that the comparison to SFT regarding the number of training examples ("TT-SI (...) trains using 68x less training samples") in line 372, Figure 1, and Figure 2 is misleading: SFT "examples" come from the training set and are a one-time cost. TT-SI "examples" come from the test/deployment stream and scale with how many predictions are served.
- The runtime comparison to SFT and the claim that "TT-SI delivers a 3.7x wall clock speed-up" in Appendix G are misleading for the same reason.
- The y-axes (accuracy) in the bar charts in Figures 1 and 2 do not start at 0. This visually inflates the differences between Base, SFT, and TT-SI.
- Some details regarding the experimental setup are not thoroughly explained in the manuscript (see questions), which limits reproducibility. (And the authors do not provide source code.)

Minor issues and typos:
- In Figure 1, it seems that "Certain" and "Uncertain" are swapped.
- In line 281, the "ij" indices are not written in subscript.
- In line 358/359 "...API-Bank,sing both..."
- In line 377, the accuracy improves from 66.37% to 66.38%, while in Figure 2, it improves from 66.37% to 68.36%
- Line 410 "ere"
- $\mathcal{D}_i'$ in lines 190 and 192 supposedly should be $\mathcal{D}_i$

**Questions:**

1. Can one expect an increase in accuracy of TT-SI w/o H in Figure 2 when increasing the number of supplied examples while keeping $K$ constant? If yes, why?
2. What are the standard deviations of the reported metrics for the five individual runs in Table 1?
3. What is majority-vote (self-consistency) in Table 1? Does it refer to the decoding strategy proposed by [1]? If yes, why is the paper not cited?
4. What is the "scale" in line 412? Does it mean the number of samples in a query-specific data set $\mathcal{D}_i$, or the number of samples in $\mathcal{D}_\mathrm{test}$, or something else?
5. How does the size of the $\mathcal{D}_i$ influence accuracy?

**References**
- [1] Xuezhi Wang et al. Self-Consistency Improves Chain of Thought Reasoning in Language Models. ICLR 2023.

---

### Note · Authors · 2025-11-19

I have read and agree with the venue's withdrawal policy on behalf of myself and my co-authors.